# OpenReview forum: "YaPO: Learnable Sparse Activation Steering Vectors for Domain Adaptation"
_ICLR.cc/2026/Conference — ICLR 2026 Conference Withdrawn Submission_

### Official Review · Reviewer_Xmej · 2025-10-26

**Soundness:** 2
**Presentation:** 2
**Contribution:** 2
**Rating:** 2
**Confidence:** 3

**Summary:**

The paper centers on activation steering, extending Bi-directional Preference Optimization (BiPO) by applying it within the sparse space of a Sparse Autoencoder. The experiments primarily target cultural adaptation, with results on Gemma 2 showing superior performance over BiPO.

**Strengths:**

1. The proposed approach is intuitive and easy to understand.


2. In cultural adaptation tasks, it achieves significant improvements over the baseline method, BiPO.

**Weaknesses:**

1. The experiments were only conducted on Gemma2-2B, and the results need to be validated on more models to demonstrate generality.

2. The baselines are limited. The paper only compares against BiPO, while there are many existing works on sparse activation steering that should be included for a more comprehensive comparison.

3. The tasks are restricted to cultural adaptation, and although the authors created their own dataset, the description of the task is vague. It is difficult to understand the actual goal of this task.

4. In Section 4.4 Generalization to Other Domains, the authors only report results on hallucination reduction without providing any details about the experimental setup, which makes this section confusing.

5. Regarding method design, the authors follow BiPO’s bidirectional training framework, but they do not specify what the target behavior and opposite behavior are for the given task. Moreover, the evaluation does not mention bidirectional steering, leaving the purpose of this design unclear.

6. Minor: Line 290 and 785 contain invalid pointers.

**Questions:**

See Weaknesses.

---

> ### Author Response · Authors · 2025-12-03
>
> We thank the reviewer for their feedback. We have substantially revised the paper to improve clarity and experimental breadth.
>
> 1. “Only Gemma-2-2B; need more models”
>
> We addressed by adding Gemma-2-9B-it experiments (cultural MCQ, cultural open-ended, general safety). See Tables 8-10 in the revised draft.
> We acknowledge that Llama/Qwen architectures remain uncovered due to compute and time constraints, and we now clearly state this.
>
> 2. “Baselines limited; need more sparse steering works”:
>
> We have expanded baselines to include:
> •	CAA and SAS alongside BiPO and YaPO, across all cultural and safety metrics.
> •	A more complete related-work discussion of sparse steering methods (including SAE-TS / SAE-SSV, ReFT, etc.), clarifying which ones we compare to empirically and which we only discuss conceptually due to compute differences.
>
> 3. “Task definition and dataset description vague”:
>
> We substantially clarified:
> •	The task goal (aligning answers toward local cultural norms while keeping utility),
> •	The construction of MCQ vs open-ended prompts,
> •	How localized vs non-localized prompts are built,
> •	Examples of target vs opposite behaviors used for bi-directional training (e.g., local vs Western answer templates),
> •	Dataset statistics and curration pipeline and we plan to release it.
>
> 4. “Generalization to other domains: setup unclear”:
>
> We now fully describe the generalization tasks evaluation with the prompts we used (similar to the ones used in BiPO) and report results for all four methods on both 2B and 9B. All broken references (Line 290, 785) and missing figures have been fixed in the revised PDF.

---

### Official Review · Reviewer_aqPy · 2025-10-29

**Soundness:** 2
**Presentation:** 3
**Contribution:** 2
**Rating:** 4
**Confidence:** 4

**Summary:**

This paper introduces YaPO (Yet Another Policy Optimization), a method for learning sparse steering vectors in the latent space of Sparse Autoencoders (SAEs) to improve LLM alignment and domain adaptation. Unlike dense steering methods such as BiPO that operate directly in activation space and suffer from neuron multi-semanticity, YaPO optimizes sparse codes using a bi-directional preference optimization objective, producing disentangled and interpretable steering directions. The authors focus on cultural alignment as a case study, curating a new multilingual dataset covering 15 cultural contexts across 5 language families with both localized and non-localized prompts to measure the explicit-implicit localization gap.

The experimental results on Gemma-2-2B demonstrate that YaPO converges significantly faster than BiPO (under 150 steps vs. 600+ steps), achieves substantial performance improvements across multiple-choice questions (+14.7% average) and open-ended generation tasks, and remains more stable throughout training. YaPO also reduces the Performance-Normalized Localization Gap (PNLG) by 27.3% while improving Robust Cultural Accuracy (RCA) by 54.3%, indicating better consistency between localized and non-localized prompts. Beyond cultural alignment, the method generalizes to other alignment tasks such as hallucination mitigation, establishing sparse steering as a scalable approach for fine-grained LLM control.

**Strengths:**

- First method to combine preference optimization with sparse steering vectors in SAE latent space, addressing limitations of both dense steering (BiPO) and static sparse methods (SAS)

- Demonstrates order-of-magnitude faster convergence and consistent performance improvements across all evaluated languages and settings

- Curates a high-quality multilingual dataset (45,354 items) with careful controls for dialect, cultural validity, and localized/non-localized variants

- Introduces PNLG and RCA metrics that appropriately measure both absolute performance and robustness to implicit cultural cues

**Weaknesses:**

- My biggest concern with this paper is the lack of baselines regarding steering with SAE. The authors did not compare against some new baselines like ReFT-r1, RePS, HyperSteer, and EasyEdit2. Since these methods also leverage SAE-based representations for steering, this omission makes it difficult to assess whether YaPO's improvements are genuinely novel.

- I am a little bit concerned about the limited model coverage. YaPO is only evaluated on Gemma-2-2B (briefly mentions Gemma-2-9B), lacking evidence of scalability to larger models or different architectures (Llama, Qwen, etc.). There are also SAEs provided for models like Pythia and Llama in SAELens, and therefore I think more experiments are reasonable and necessary.

- The interpretability claims are relatively unclear. While claiming "interpretable" steering, the paper lacks systematic analysis of what individual sparse features encode or how they differ from BiPO's dense features. Some automatic annotations with feature activations analyzed by LLMs could be a good supplementary material to strengthen these claims.

- The cultural dataset focuses on specific countries but may not capture within-country diversity; Western "control" answers may introduce bias. The authors could consider using more datasets like those used in CAA and Axbench to demonstrate broader applicability.

**Questions:**

1. Why did the author not compare against recent SAE-based steering methods like ReFT-r1, RePS, HyperSteer, and EasyEdit2?

2. Can the author provide results on larger models (e.g., Gemma-2-27B, Llama-3) or different architectures (Qwen, Pythia) to demonstrate scalability?

3. What specific concepts do the learned sparse features encode, and how do they differ from BiPO's dense features?

4. Have the author tested YaPO on established cultural alignment benchmarks like those used in CAA and Axbench to validate broader applicability?

5. How does the author method handle within-country cultural variations, given that your dataset focuses on country-level differences?

---

> ### Author Response · Authors · 2025-12-03
>
> We thank the reviewer for the constructive comments and the encouraging rating. We have addressed the weaknesses regarding baselines, model coverage, interpretability and dataset.
>
> 1. “Missing baselines: ReFT-r1, RePS, HyperSteer, EasyEdit”:
>
> We appreciate this pointer. These methods indeed operate on structured representations, but they are weight-modification / finetuning / editing approaches with different training costs and operational regimes than test-time activation steering.
> To clarify our approach, we first distinguish YaPO as being more of test-time approach that is reference free and requires only DPO like preference data. On the other hand:
> •	ReFT & RePS: "We categorize these as parameter-efficient fine-tuning (PEFT) or supervised dictionary learning methods. They require significantly higher training compute to learn adaptors or dictionaries compared to YaPO's lightweight vector optimization."
> •	HyperSteer: This method requires training a separate hypernetwork (a transformer) to generate vectors. This introduces substantial architectural overhead and training cost compared to YaPO's direct optimization.
> •	SAE-SSV: This is a relevant test-time steering baseline. However, it relies on training linear classifiers (probes) on labeled data to find directions. YaPO uses a preference-optimization (DPO like) objective, which is distinct and closer to BiPO instead.
> •	SAE-TS: “SAE-TS uses a pretrained SAE to quantify how steering vectors affect sparse features, then learns steering vectors that target desired features while minimizing collateral changes.”
> •	EasyEdit2: “EasyEdit2 is a general steering/editing framework that unifies prompt-based, activation-based, and (future) decoding-based interventions, providing tooling and evaluation rather than a single steering algorithm.”
>
> We now explicitly mention ReFT-r1, RePS, HyperSteer, EasyEdit in the related work and explain that our focus is on lightweight, test-time steering methods (CAA, SAS, BiPO, YaPO). Full comparison to the above would be unfair and require multiple finetuning runs per model, which is beyond our compute budget in this work.
> We believe adding CAA and SAS closes the gap among activation-level steering baselines, while acknowledging that a broader comparison to editing / finetuning methods is an important next step.
>
> 2. “Limited model coverage; need larger models / different architectures”:
>
> As discussed above, we have now added Gemma-2-9B-it experiments. This shows that:
> •	All steering methods still help a stronger base model.
> •	Differences between methods narrow on MCQ, but sparse learned steering (SAS/YaPO) remains strong for open-ended cultural prompts.
> We agree that experiments on Llama-3, Qwen, Pythia, etc. using SAELens SAEs would be valuable; however, this would require training or hosting SAEs and multiple steering runs for each, which is currently beyond our compute and time budget.
>
> 3. “Interpretability unclear”:
>
> We revise the interpretability claims. We now position YaPO as operating in an interpretable basis (SAE features) rather than providing a full interpretation of each feature.
>
> 4. “Dataset diversity and external benchmarks”:
>
> •	We clarify that our dataset is country-level, not intended to capture the full within-country diversity, and we now explicitly state this limitation.
> •	We discuss that Western control answers may introduce bias and frame this dataset as a first step toward structured cultural evaluation rather than a complete solution.
> •	We evaluated all four methods (CAA, SAS, BiPO, YaPO) on general safety tasks (hallucination, wealth-seeking, jailbreak, power-seeking) included in BiPO, SAS and CAA. This directly addresses the “limited benchmarks” concern.

---

### Official Review · Reviewer_AEFn · 2025-10-31

**Soundness:** 2
**Presentation:** 2
**Contribution:** 2
**Rating:** 2
**Confidence:** 4

**Summary:**

In this paper, the authors proposed YaPO (Yet Another Policy Optimization) which is an optimization-based approach to find steering vectors with the help of a pretrained SAE (sparse auto-encoder) of the target model. Specifically, YaPO trains the steering vector in a very similar way to BiPO but moves the steering vector from the model's hidden representation to the sparse features' activation, in hope that it will mitigate the multisemanticity issues in BiPO steering vectors. YaPO is only tested on Gemma-2 (and only results on the 2B variant are disclosed) with the off-the-shelf SAE Gemma Scope and primarily for a cultural localization task with a dataset the authors collected on their own, where BiPO yields noticeable improvement over both BiPO and the model w/o steering. The authors also shared some information on YaPO's performance in steering against hallucination which also outperforms BiPO and the model w/o steering.

**Strengths:**

+ It is good to see more research on joining SAE and steering vectors.
+ The cultural localization problem the authors put forward and gathered a dataset for is an interesting problem and can be a good addition to existing tasks for benchmarking model behavior manipulation.

**Weaknesses:**

+ The idea of bridging SAE and steering vectors are not exactly new. For instance [1] and [2] both have investigated how sparsity/monosementicity helps regularizes representation steering. In a way, YaPO can be considered merely using BiPO to achieve [2].
+ While BiPO is a very good paper to base on, using it as the only baseline is inadequate, given that there are existing works that shared the same design as mentioned above.
+ The experiments are also limited.
    + Gemma is the only model being evaluated meaning that YaPO has never been evaluated on a model (mostly Llama, vicuna and their variants) that BiPO was originally tested on, even though there is off-the-shelf SAEs for them like Llama Scope as well. The performance on a single model is not as convincing as that across multiple models.
    + The major experiments are conducted for the cultural localization task with datasets the authors built on their own without sharing a single example except for the 2 phrases in section 3.1. The authors also included some results about hallucination without any details other than the scores. BiPO was said to perform even worse than no steering which contradicts its reported performances on other models. The authors also fail to demonstrate if YaPO would undermine general utility of the models with e.g. MMLU benchmark.

    If the authors do want to closely follow BiPO, they are suggested to at least do the experiments that BiPO has done with the exact settings. Simply be comparing Gemma Scope and Llama Scope it is not hard to find that they are very different in terms of how the sparse features look like, so it is possible that YaPO is only better for Gemma based models on very specific tasks.
+ YaPO is said to be more efficient to train but that claim assumes that there is a pretrained SAE available for whichever model one want to use YaPO on. However in reality, training an SAE is actually way more consuming than either BiPO or YaPO and one cannot always expect a pretrained SAE to be readily available. People are interested in joining steering vectors and SAEs because they, to some extent, comprise a dual formulation of each other — one building up each single feature vectors bottom up from preference datasets, the other finds all possible feature vectors top down in an unsupervised fashion. So having a pretrained SAE at hand naturally gives YaPO a leverage by having the majority of the work done already so it is not surprising at all YaPO optimization could be faster.

1. Chalnev, Sviatoslav, Matthew Siu, and Arthur Conmy. "Improving steering vectors by targeting sparse autoencoder features." arXiv preprint1 arXiv:2411.02193 (2024).
2. He, Zirui, et al. "SAE-SSV: Supervised Steering in Sparse Representation Spaces for Reliable Control of Language Models." arXiv preprint arXiv:2505.16188 (2025).

**Questions:**

Please refer to the weakness for the concerns I have about this paper. Here I am just listing a few questions to facilitate the understanding of my concerns.
+ Could it be 2B model is too small for the tasks?
+ Could it be Gemma's SAE being different from Llama's?
+ Did you redo the layer selection and hyperparameter search for BiPO?
+ How does YaPO compare to other approaches to enhance steering vectors with SAE guidance?
+ What if there is no off-the-shelf SAE? Can you train a sparse steering vector without a pretrained SAE?

---

> ### Author Response · Authors · 2025-12-03
>
> We thank the reviewer for the constructive comments. We have addressed the weaknesses regarding baselines and model coverage.
>
> 1. “Idea not new; relation to SAE-TS / SAE-SSV”:
>
> We fully agree that bridging SAEs and steering is an active area and should be discussed more thoroughly. We now:
> •	Explicitly cite and discuss
> o	Chalnev et al., Improving steering vectors by targeting sparse autoencoder features (SAE-TS)
> o	He et al., SAE-SSV: Supervised Steering in Sparse Representation Spaces.
> •	Clarify that these works use supervised objectives (class labels, signed scores) on sparse codes, whereas YaPO adopts a bi-directional preference optimization objective over SAE activations (similar to BiPO) and is tailored to settings where we have preference-style comparisons rather than explicit labels.
> Therefore the novelty lies in how we train sparse steering (BiPO-style preference optimization in SAE space), not just the observation that sparse features are useful.
>
> 2. “BiPO as sole baseline is inadequate”
>
> We agree more baselines were needed and we have added more comparisons for more fairness:
> •	CAA (contrastive activation addition)
> •	SAS (sparse activation steering with static sparse directions)
> and evaluate all four methods (CAA, SAS, BiPO, YaPO) on:
> •	MCQ RCA + PNLG,
> •	open-ended RCA + PNLG, and
> •	general safety tasks (hallucination, wealth-seeking, jailbreak, power-seeking)
> This directly addresses the “limited baselines” concern in the activation-engineering / sparse-steering family.
> We would like to mention that we do not compare to methods like ReFT-r1, RePS, HyperSteer, EasyEdit, etc., which are weight-update / finetuning / editing methods with substantially different compute and training costs; we instead focus on methods that can be applied at test time. We now clarify this scope difference in the related-work section.
>
> 3. “Experiments limited to Gemma-2-2B; no Llama / vicuna; no MMLU”:
>
> •	We now add Gemma-2-9B-it experiments (MCQ, open-ended, safety) to demonstrate that our conclusions hold at a larger scale.
> •	We agree that additional architectures (Llama, Qwen, etc.) would strengthen the paper. Nonetheless Using Llama-scope 8B and running BiPO/YaPO on multiple cultural and safety benchmarks requires substantial GPU time that we do not currently have.
> Regarding MMLU: rather than re-running a full general-knowledge suite, we used BiPO’s general safety benchmarks (hallucination / power-seeking / etc.) as a proxy to test whether steering breaks general behavior. We now describe those tasks and the scoring in more detail and show that none of the steering methods harms aggregate performance, and CAA/SAS/YaPO often improve it.
>
> 4. “Cultural dataset vague; hallucination results unclear; BiPO layer selection”
>
> •	Dataset: the appendix now contain more details (counts, sampling, templates, examples, and curation pipeline).
> •	General tasks:
> o	We now specify the benchmark source (BiPO’s released evaluation suite), the scoring and the number of prompts.
> o	We also report CAA and SAS results here, not just BiPO and YaPO, and show that all methods offer modest but consistent gains.
> •	BiPO tuning details:
> o	We clarify that for BiPO we re-used the best layer range from preliminary sweeps and performed a small grid search over the steering weight $\lambda$ (details added in the appendix).
>
> 5. “Efficiency claims with pretrained SAEs”
>
> We agree that training an SAE is expensive. In the revised text we:
> •	Clarify that our efficiency claim is conditional on having an SAE already available (e.g., GemmaScope, LlamaScope). In that regime, optimizing in sparse space converges in far fewer steps than BiPO on dense activations.
> •	Explicitly state that training SAEs themselves is out of scope and that YaPO should be viewed as a method that leverages (rather than replaces) the SAE investment.
> Regarding “what if no SAE is available?”: we discuss this as an interesting direction. One could learn task-specific small SAEs or low-rank sparse projections, and we leave it for future work.

---

### Official Review · Reviewer_Fp8b · 2025-11-01

**Soundness:** 1
**Presentation:** 2
**Contribution:** 1
**Rating:** 2
**Confidence:** 4

**Summary:**

Instead of learning a steering vector, they propose YaPO, where they instead learn to steer sparse features of a SAE. They show this on a cultural benchmark that they curate, and show that the method converges faster than and outperforms BiPO on that benchmark, as well as a hallucination benchmark.

**Strengths:**

* The method converges much faster than BiPO and outpeforms BiPO in the cultural benchmark.
* They also perform their method on BiPO's benchmarks, but only on the hallucinations dataset (which they note in their Limitations section)

**Weaknesses:**

* The work was done only on a single 2B model. The 9B variation was mentioned once in Limitations with no further details in the main body or Appendix.
* The paper claims to produce more interpretable steering directions, but fails to do any work on interpreting the steering direction. They note that this is "beyond the scope of this paper" in the Limitations, but I disagree, as merely using the sparse autoencoder feature basis is not sufficient to make things more interpretable.
* While the dataset/benchmark is claimed as a main contribution, there is barely any information about it in the main body.

**Questions:**

In addition to the three weaknesses mentioned, I have the following questions:
* How is this work different from [SAE TS](https://arxiv.org/abs/2411.02193)? (I note that SAE TS is not peer-reviewed and was not factored into my accept/reject decision, but nevertheless the paper should be cited and discussed as it predates this work by more than a year).
* Is there any particular reason you choose to report the "Egypt" evaluation performance only for Figure 1? Is it possible to instead report the average difference over all categories between YaPO and BiPO across training epochs? What does that look like?

I also have the following feedback:
* **There are hallucinated citation authors**, like "Steering llama 2 via contrastive activation addition" being attributed to a "Nathan Rimsky". "Steering Language Models With Activation Engineering" also has hallucinated author names.
* Appendix B is incomplete, especially its last paragraph which seems to be a broken transcript.
* Line 290: Appendix reference missing
* L418: Typo immediately after Activation Engineering (the full-stop).
* L785: Figure missing
* L302 Incomplete line "we observe that the performance improvement is stable and consistent throughout the epochs for YaPO while BiPO"...

---

> ### Author Response · Authors · 2025-12-03
>
> We thank the reviewer for their valuable feedback. We have revised the paper to address your concerns regarding model scale, interpretability, and the dataset.
>
> 1. “Single 2B model, no 9B details”:
> We agree scalability is important. We have now added Gemma-2-9B-it experiments for
> o	MCQ cultural evaluation
> o	open-ended cultural prompts
> o	general safety / alignment tasks
> Across these, all steering methods improve over the base 9B model, and BiPO/SAS/YaPO provide similar gains on MCQ, while SAS/YaPO tend to dominate on open-ended cultural prompts. The main conclusions is that BiPO is strong but sparse learned steering remains competitive and often better on long-form tasks hold at 9B.
> For clarity of presentation, we left the main body for 2B and did add the scalability in Appendix E.
>
> 2. “No interpretability analysis despite the ‘interpretable’ claim”:
> We agree that “interpretable” should be used carefully and would like to highlight that our main contribution is an optimization procedure over a sparse latent basis, not the interpretability study itself.
> We have softened the claim in the main text: we now emphasize that we operate in a structured sparse feature space (SAE codes), which is more amenable to interpretability than dense activations, rather than claiming fully interpreted features.
> We believe this aligns the claims with what we actually show.
>
> 3. “Dataset is a main contribution but poorly described”:
> We have expanded the dataset curration pipeline with more description:
> •	A table with item counts per language and country (5 languages × 15 countries).
> •	Clear explanation of localized vs non-localized prompt templates, the Western control answer, and how we construct MCQ vs open-ended tasks.
> •	More textual examples of prompts and answers for several languages.
> •	Clarification that the full dataset (with templates and annotation guidelines) will be released.
> We hope this addresses the “barely any information” concern. The details were deffered to the Appendix for space and clarity of presentation.
>
> 4. “Difference from SAE-TS; why only Egypt in Fig. 1; misc. issues”:
> •	Relation to SAE-TS / Chalnev et al.
> We now explicitly cite and discuss SAE-TS and SAE-SSV (He et al.) in the related work. We would like to emphasise that conceptually, SAE-TS and SAE-SSV are closely related sparse methods, but they also rely on supervised labels over SAE codes. Implementing their pipelines on our cultural benchmark would require an additional round of SAE-feature annotation and steering-specific training, which is beyond our current compute/time budget. We instead compare to their direct activation-engineering ancestors (CAA, SAS) and to BiPO. On the other hand, YaPO optimizes preference-based objectives (DPO-style) over sparse codes, including bi-directional preferences and per-token credit assignment. In other words, SAE-TS is closer to supervised steering, while YaPO is DPO-style preference optimization but in SAE space.
> •	Why only Egypt in Figure 1?
> We would like to mention that there is no particular reason for chosing Egypt and that originally the paper has a figure showing the performance across different countries. We showed Egypt only to avoid clutter.
> •	Citations and formatting issues:
> We fixed all hallucinated author names, missing references, and broken sentences/figures mentioned (e.g., missing Fig., incomplete line around L302, broken appendix paragraph, punctuation after “Activation Engineering”).

---

### Note · Authors · 2026-01-06

**Comment:**

Dear Reviewers,
We sincerely appreciate your thoughtful reviews and the time you dedicated to evaluating our work. After consideration, we have decided to withdraw the submission. Thank you again for your constructive feedback.
All the bests,
Abdelaziz Bounhar

**Withdrawal Confirmation:**

I have read and agree with the venue's withdrawal policy on behalf of myself and my co-authors.